# Rotation-Scale Equivariant Steerable Filters

**Yilong Yang**                                                    Yilong.Yang@soton.ac.uk
**Srinandan Dasmahapatra**                                         sd@ecs.soton.ac.uk
**Sasan Mahmoodi**                                                 sm3@ecs.soton.ac.uk
*University of Southampton, University Road, Southampton, SO17 1BJ, United Kingdom*

**Editors:** Accepted for publication at MIDL 2023

## Abstract

Incorporating either rotation equivariance or scale equivariance into CNNs has proved to be effective in improving models' generalization performance. However, jointly integrating rotation and scale equivariance into CNNs has not been widely explored. Digital histology imaging of biopsy tissue can be captured at arbitrary orientation and magnification and stored at different resolutions, resulting in cells appearing in different scales. When conventional CNNs are applied to histopathology image analysis, the generalization performance of models is limited because 1) a part of the parameters of filters are trained to fit rotation transformation, thus decreasing the capability of learning other discriminative features; 2) fixed-size filters trained on images at a given scale fail to generalize to those at different scales. To deal with these issues, we propose the Rotation-Scale Equivariant Steerable Filter (RSESF), which incorporates steerable filters and scale-space theory. The RSESF contains copies of filters that are linear combinations of Gaussian filters, whose direction is controlled by directional derivatives and whose scale parameters are trainable but constrained to span disjoint scales in successive layers of the network. Extensive experiments on two gland segmentation datasets demonstrate that our method outperforms other approaches, with much fewer trainable parameters and fewer GPU resources required. The source code is available at: https://github.com/ynulonger/RSESF.

**Keywords:** Scale, Rotation, Equivariant, Segmentation

## 1. Introduction

Recent work has shown that exploiting rotation and scale symmetries in convolutional neural networks can improve models' generalization performance (Cohen and Welling, 2016) and sample efficiency (Linmans et al., 2018; Yang et al., 2022). Domains that benefit most from exploiting both symmetries are those where the image itself lacks canonical orientation and scale, and where the number of samples is limited. These are features of digital histopathology images, where localized patterns can appear in any orientation, and the size of cells and tissues covary with different choices of objective magnifications used to digitalize specimen slides. In medical image-related tasks (which have very limited samples), one can use offline data augmentation for better generalization, but this is at the cost of increasing the number of training samples. Previous work (Linmans et al., 2018; Veeling et al., 2018; Graham et al., 2019; Bekkers et al., 2018; Graham et al., 2020) shows that incorporating rotation equivariance into CNNs leads to better performance on the classification, detection, and segmentation of histopathology images. Yang et al. (2022) demonstrates that introducing scale equivariance into CNNs improves models' segmentation performance when tested on histopathology images that are presented in unseen scales.

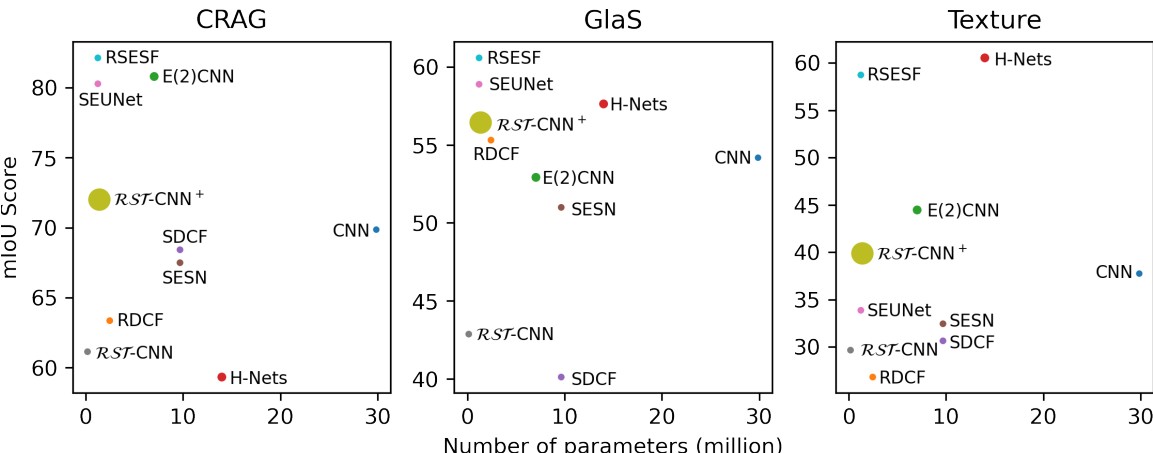

Figure 1: **Model Size vs. Segmentation Performance vs. GPU Requirement.**
Models evaluated on out-of-distribution setting. Our RSESF outperforms others
on CRAG and GlaS datasets, achieving competitive results on the texture dataset.
Notably, the RSESF achieves state-of-the-art mIoU but is much smaller and more
GPU efficient. The larger the diameter of the marker, the larger the GPU memory
required for model training. Details are in Table 1.

However, all of these works either consider rotation symmetry or scale symmetry, which did
not fully extract CNNs' potential for joint rotation-scale equivariance.

In this paper, we introduce the Rotation-Scale Equivariant Steerable Filter (RSESF),
which utilizes filter steerability and Gaussian scale-space theory to parameterize convolu-
tional filters, resulting in an equivariant layer that is stable to rotation and scale variations.
Figure 1 summarizes the generalization capability of models on three datasets, where our
RSESF demonstrates superior generalization performance on two histopathology datasets
but uses fewer parameters and demands less GPU memory on training. More specifically,
the equivariance is achieved by using a Gaussian derivative filter basis that is jointly con-
trolled by a scale parameter $\sigma$ along with an orientation parameter $\theta$ to manipulate the
receptive field size and rotation angle of convolutional filters. Low GPU memory usage
of the training process is achieved by only training one orientation of the filter; the other
orientations are only generated at inference when required.

## 2. Related Work

**Rotation Equivariant CNNs.** Cohen and Welling (2016) propose group equivariant
CNNs (G-CNNs) where rotation and reflection symmetries are embedded into G-convolution.
The restriction to 90° rotations was lifted in subsequent work, and (Worrall et al., 2017;
Weiler et al., 2018; Cheng et al., 2019; Weiler and Cesa, 2019) used the concept of steerable
filters to construct filters with either discrete or continuous rotations. For example, Cheng
et al. (2019) propose the RDCF, which decomposes filters over joint steerable bases across
spatial translations and discrete group symmetry simultaneously. Worrall et al. (2017) in-

troduce H-Nets (Harmonic Networks) that achieve continuous rotational equivariance by building filters out of the family of circular harmonics.

**Scale Equivariant CNN.** In the pre-deep learning era, Gaussian scale-space theory (Lindeberg, 1994) was widely used for multi-scale image representation. Recently, (Lindeberg, 2022; Yang et al., 2022) have parameterized convolutional filters as a linear combination of Gaussian derivative filters with different scales, building neural networks robust to scale variations on image classification and segmentation tasks. Other works that adopt symmetry principles to establish scale-convolution include (Sosnovik et al., 2021; Worrall and Welling, 2019; Sosnovik et al., 2020; Zhu et al., 2022). In (Sosnovik et al., 2020), the authors propose Scale-Equivariant Steerable Networks (SESN), where filters are parameterized by a trainable linear combination of pre-calculated Hermite basis functions. Similarly, (Zhu et al., 2022) propose Scale Decomposed Convolutional Filters (SDCF) that decompose the convolutional filters under two pre-determined separable bases and truncate the expansion to low-frequency components.

**Roto-Scale-Translation Equivariant CNN.** All of the aforementioned literature encodes the rotation and scale equivariance properties into CNNs separately. The $\mathcal{RST}$-CNN (Gao et al., 2022) is the first attempt to simultaneously incorporate translation, rotation, and scaling symmetry into convolutional layers. However, it is only evaluated on image classification tasks on small and simple datasets (rotation and scale augmented MNIST (LeCun, 1998), Fashion-MNIST (Xiao et al., 2017), and STL-10 (Coates et al., 2011)). In image classification tasks invariance to global transformations of scale and rotation is key to predictive accuracy, while the equivariance to scaling and rotation is more important for segmentation. Furthermore, the training of $\mathcal{RST}$-CNN requires the GPU memory several times more than what is needed by conventional CNNs, which greatly limits its applicability.

## 3. Methodology

### 3.1. Rotation-Scale Steerable Gaussian Derivative Filter Basis

The 1D Gaussian filter at scale $\sigma$ is written as $G(x; x_0, \sigma) = \frac{1}{\sigma\sqrt{2\pi}} e^{-\frac{(x-x_0)^2}{2\sigma^2}}$ which can be extended to 2D isotropic Gaussian filters as $G(x, y; x_0, y_0, \sigma) = G(x; x_0, \sigma)G(y; y_0, \sigma)$. We will drop the centres $x_0, y_0$ to simplify notation. In (Pintea et al., 2021; Lindeberg, 2022; Yang et al., 2022), the authors linearly combine the 2D Gaussian derivatives,

$$G^{i,j}(x, y; \sigma) = \frac{\partial^{i+j} G(x, y; \sigma)}{\partial x^i \partial y^j} = \frac{\partial^i G(x; \sigma)}{\partial x^i} \frac{\partial^j G(y; \sigma)}{\partial y^j}, \; i, j \geq 0, \; i + j \leq N \qquad (1)$$

to construct filters. Order $N$ filters refer to the highest order of derivative used in equation (1). These basis elements capture variations along the $x$ or $y$ directions. We define filter basis elements $G_\theta^1(x, y; \sigma)$ rotated by angle $\theta$:

$$\begin{aligned} G_\theta^1(x, y; \sigma) &= \cos\theta \frac{\partial G(x, y; \sigma)}{\partial x} + \sin\theta \frac{\partial G(x, y; \sigma)}{\partial y} = (\cos\theta \partial_x + \sin\theta \partial_y) G(x, y; \sigma) \\ &= \cos\theta G_0^1(x, y; \sigma) + \sin\theta G_{\frac{\pi}{2}}^1(x, y; \sigma) \end{aligned} \qquad (2)$$

with $\cos\theta, \sin\theta$ as interpolation functions as in steering theorems (Freeman et al., 1991). Using equation (2), we can simultaneously control the size and orientation of convolutional

filters by manipulating $\sigma$ and $\theta$ parameters of their filter basis. The benefit of using a steerable filter is that it avoids the interpolation artifacts produced by directly rotating the filter. The $N = 2$ filters are $(\cos\theta\partial_x + \sin\theta\partial_y)^2 G(x, y; \sigma)$ as shown in Appendix A.

### 3.2. Filter Construction.

We denote the filter basis rotated by $\theta$ as $G_\theta^{i,j}(x, y; \sigma)$. Then we parameterize the proposed RSESF filter as a linear combination of directional Gaussian derivative filters:

$$F_k^l(c^l, x, y, \sigma_k^l, \theta_r; c^{l-1}) = \sum_{\substack{i,j\geq 0}}^{i+j\leq N} \alpha_{i,j,c_l,c^{l-1}}^l G_{\theta_r}^{i,j}(x, y; \sigma_k^l), \ 1 \leq k \leq \gamma, \ \theta_r = \frac{2\pi r}{R}, 1 \leq r \leq R$$

(3)

where $l \geq 1$ is the layer index, $c^{l-1}$ and $c^l$ are the channel indices of the input and output of the $l^{th}$ layer, $k$ is the index for the $\gamma$ scales and $r$ for the $R$ orientations in a layer. The expansion coefficients $\alpha_{i,j,c^l,c^{l-1}}^l \in \mathbb{R}$ and scale parameter $\sigma_k^l$ are learnable. The filter $F_k^l$ constructed from equation (3) is of the dimension $[C^l, C^{l-1}, R, h_k^l, w_k^l]$, which has $R$ rotation channels. $(h_k^l, w_k^l)$ denotes the spatial size of the filter, which is controlled by the scale parameter $\sigma_k^l$, i.e., $h_k^l = w_k^l = 2\lceil 2.5\sigma_k^l\rceil + 1$. We create $\gamma$ groups of filters $[F_1^l, \cdots, F_\gamma^l]$, with each group sharing the same expansion coefficients $\alpha_{i,j,c^l,c^{l-1}}^l$, but with different scale factors $\sigma_k^l$s to ensure scale equivariance. Within each scale, $R$ rotation channels at angular resolution $\frac{2\pi}{R}$ implement discrete rotation equivariant filters. Figure 3 in section B illustrates the construction of the filter. We then describe feature extraction across scales in parallel.

### 3.3. Equivariant Convolution

For the first layer, we convolve the image with each group of filters in parallel. For channel $c^1$, $1 \leq c^1 \leq C^1$ at scale $k$ and rotation $r$, the convolution $f_k^1 = F_k^1 \star f^0$

$$f_{k,c^1,r}^1(x, y) = \sum_{c^0=\mathbf{r,g,b}} \sum_{x_0,y_0} F_k^1(c^1, x - x_0, y - y_0, \sigma_k^1, \theta_r; c^0) f_{c^0}^0(x_0, y_0)$$

$$= \sum_{\substack{i,j\geq 0}}^{i+j\leq N} \sum_{c^0=\mathbf{r,g,b}} \sum_{x_0,y_0} \alpha_{i,j,c^1,c^0}^1 G_{\theta_r}^i(x; x_0, \sigma_k^1) G_{\theta_r}^j(y; y_0, \sigma_k^1) f_{c^0}^0(x_0, y_0)$$

(4)

gives the output $f_k^1 \in \mathbb{R}^{C^1 \times R \times H \times W}$ for each of the $\gamma$ scales.

For subsequent layers ($l \geq 2$), the feature map from each scale group is only passed to the corresponding scale also indexed by $k$, i.e., $f_k^l = F_k^l \star f_k^{l-1}$. Inside each scale group, for each output channel, feature maps of multiple orientations from the previous layer are independently convolved with multi-orientated filters and then summed over orientation channels:

$$f_{k,c,r''}^l = \sum_{d=1}^{C^{l-1}} \sum_{r'=1}^{R} F_{k,c,d,r'}^l * f_{k,d,r''}^{l-1}, \quad c \in \{1, \cdots, C^l\}, k \in \{1, \cdots, \gamma\},$$

(5)

where $r'$ and $r''$ are the indices of rotation channel for the filter and feature maps, respectively. The sum over orientation channels renders transformations between hidden layers

invariant to rotations. Note, features from other orientation channels $r \in \{1, \cdots, R\}$, $r \neq r''$ of the input $f_{k,c}^{l-1}$ are not involved in the calculation of $f_{k,c,r''}^{l}$, which means no orientation information is mixed through convolution. This gives the constructed network flexibility to adopt different numbers of rotation channels between the training and testing phases. This would thus enable the network to reduce the demand on GPU memory required for training while maintaining performance during inference. We will describe this benefit later.

### 3.4. Model Training

**Decoupled convolution between rotation channels enables memory efficient training.** Within each scale group, equation (4) indicates that the input image is convolved with $R$ copies of rotated filters, separately. Equation (5) shows that the input of each orientation channel of hidden layers is also individually convolved with rotated filters to generate feature maps. Since there is no inter-rotation interaction between rotation channels, the information flow is independent across rotation channels. Therefore we are allowed to train the network within only one rotation channel, but then other $R$-1 rotation channels after training are created to reduce the model's orientation sensitivity as needed. The filters of newly created rotation channels are guaranteed to be in the same shape and scale as the trained one, as the expansion coefficients $\alpha$ are shared in rotation and scale dimension. The reduction of the number of rotation channels reduces GPU memory consumption by a factor of $R$ during training, as other $R$-1 feature maps do not need to be stored in the memory in the back-propagation calculations. This translates into a two-fold advantage: firstly, a larger number of filters can be used, thus increasing the feature representation capability of the network; secondly, RSESF can be trained in GPU resource-limited settings, thus greatly increasing its applicability. Moreover, the GPU memory released from the rotation channel can compensate for the memory needed for using a larger batch-size, to achieve more stable training. Setting different number of rotation channels $R$ during inference is discussed in Appendix F.2.

**Parallel Training between Scales.** We adopt the strategy proposed in (Yang et al., 2022) for simultaneously training filters that are at different scales. The $\sigma_k^l$s remain in disjoint intervals, guaranteeing scale equivariance,

$$\sigma_k^l(x) = \frac{a_k^l - b_k^l}{2} \tanh x + \frac{a_k^l + b_k^l}{2}, \ a_k^l > b_k^l > 0, \ a_{k+1}^l > a_k^l, \ b_{k+1}^l = a_k^l, \qquad (6)$$

where $a_k^l$ and $b_k^l$ are upper and lower bounds for the scale parameters of the filters at the $l^{th}$ layer and the $k^{th}$ group. $x$ is a trainable real variable. At the last layer (the $L^{th}$ layer) of the neural network, for a $K$-class segmentation task, a $K \times C^L \times 1 \times 1$ convolutional filter is used to squeeze the feature map $f_k^L \in \mathbb{R}^{C^L \times 1 \times H \times W}$ (the 1 in the second dimension means we only use 1 rotation channel for training) into $f_k \in \mathbb{R}^{K \times H \times W}$, followed by a softmax function that maps the $f_k$ to $K$ probability maps $\hat{y}_{k,c} \in \mathbb{R}^{H \times W}$, $c \in \{1, \cdots, K\}$, which is then used to calculate the combined cross-entropy loss per-pixel

$$\mathcal{L} = -\sum_{c=1}^{K} \sum_{k=1}^{\gamma} \widetilde{\eta}_k y \log(\hat{y}_{k,c}), \ \widetilde{\eta}_k = \frac{\eta_k}{2} + \frac{1}{2\gamma}, \ 0 \leq \eta_k \leq 1, \ \sum_{k=1}^{\gamma} \eta_k = 1, \ \sum_{k=1}^{\gamma} \widetilde{\eta}_k = 1. \quad (7)$$

where $\widetilde{\eta}_k$ is a trainable rectified weighting factor to characterize the importance of the $k^{th}$ scale. $\widetilde{\eta}_k$ is bounded in $(\frac{1}{2\gamma}, \frac{\gamma+1}{2\gamma})$, which ensures that each scale contributes to the training.

Note that for the same network, the feature map $f_k^L$ is of the dimension $[C^L \times 1 \times H \times W]$ for training, but is of the dimension $[C^L \times R \times H \times W]$ in the inference phase. Therefore, in the inference phase, we max-pool the $f_k^L$ over the rotation channel to squeeze it into $(f_k^L)^{\max} = \max_{1 \le r \le R} f_{k,r}^L$, $(f_k^L)^{\max} \in \mathbb{R}^{C^L \times 1 \times H \times W}$, so $(f_k^L)^{\max}$ can be further squeezed by the following $1 \times 1$ convolutional layer. Other ways of dimension reduction are explored and compared in Appendix F.1.

## 4. Experiments and Results

### 4.1. Datasets and Compared Methods

We use the Gland Segmentation (GlaS) datatset (Sirinukunwattana et al., 2017), the colorectal adenocarcinoma gland (CRAG) dataset (Awan et al., 2017), and a synthetic texture mosaic dataset for our evaluation. More details about datasets can be found in Appendix C. We use the UNet architecture as a backbone and adopt the norm convolutional layer as well as other types of equivariant convolution layers (including RDCF (Cheng et al., 2019), E(2)CNN (Weiler and Cesa, 2019), H-Nets (Worrall et al., 2017), SDCF (Zhu et al., 2022), SESN (Sosnovik et al., 2020), SEUNet (Yang et al., 2022), $\mathcal{RST}$-CNN (Gao et al., 2022), the proposed RSESF) to generate 10 models. More details about model settings and implementation can be found in Appendix D.

Table 1: Model comparison for histopathology datasets in terms of the number of parameters, GPU memory required for training, and mIoU for ID and OOD settings. Columns $N_f$, $\gamma$, $R$ denote the number of filters, scale channels, and rotation channels in the first layer of the UNet, respectively. Note that for RSESF, "$R = 1, 8$" denotes that only 1 rotation channel is used during training but 8 rotation channels are used during testing. The total number of channels is matched between the $\mathcal{RST}$-CNN and RSESF, and the number of trainable parameters is roughly matched between the $\mathcal{RST}$-CNN$^+$ and the RSESF. A $*$ on the data denotes the presence of rotation and scale augmentation during training.

| Filter Type | Params (M) | GPU (GB) | $N_f$ | $\gamma$ | $R$ | ID Testing GlaS* | ID Testing CRAG* | OOD Testing GlaS | OOD Testing CRAG |
|---|---|---|---|---|---|---|---|---|---|
| CNN | 29.85 | 4.48 | 64 | 1 | 1 | 75.65 | 75.20 | 54.19 | 69.86 |
| RDCF | 2.42 | 5.07 | 8 | 1 | 8 | 84.38 | 86.08 | 55.33 | 63.38 |
| E(2)CNN | 7.03 | 7.65 | 8 | 1 | 8 | **84.19** | **87.91** | 52.92 | 80.79 |
| H-Nets | 13.98 | 7.65 | 64 | 1 | 1 | 72.47 | 74.44 | 57.63 | 59.34 |
| SDCF | 9.66 | 5.09 | 16 | 4 | 1 | 83.43 | 82.75 | 40.13 | 68.43 |
| SESN | 9.66 | 5.06 | 16 | 4 | 1 | 81.77 | 83.85 | 51.00 | 67.51 |
| SEUNet | 1.22 | 5.05 | 16 | 4 | 1 | 82.50 | 84.38 | 58.91 | 80.30 |
| $\mathcal{RST}$-CNN | 0.15 | 5.06 | 2 | 4 | 8 | 77.76 | 78.37 | 42.90 | 61.15 |
| $\mathcal{RST}$-CNN$^+$ | 1.36 | 15.12 | 6 | 4 | 8 | 84.08 | 85.09 | 56.43 | 72.02 |
| RSESF | 1.22 | 5.05 | 16 | 4 | 1, 8 | 83.10 | 85.00 | **60.60** | **82.15** |

## 4.2. Results

**Evaluation regime.** Two criteria are used to evaluate models' performance. 1) In-Distribution (ID) test, i.e., the training set and test set are randomly rotated (by an angle uniformly distributed on $[0, 2\pi]$) and re-scaled (by a factor uniformly distributed on [0.5, 2]). 2) Out-Of-Distribution (OOD) test, all models are trained *without* rotation and scale augmentation, but test images are randomly rotated and re-scaled. Table 1 summarizes the comparison between models in terms of segmentation performance, the number of trainable parameters, the amount of GPU memory required for training, and the training speed.

**ID testing.** For ID testing, Table 1 shows that RSESF can extract useful representations for segmentation, even with much fewer parameters. In detail, RSESF outperforms CNN, but its number of parameters is just 4.21% of CNN. The mIoU score of E(2)CNN is 1.02 and 1.34 points higher than RSESF, but the number of its parameters is 5.76 times that of RSESF. Other scale equivariant and rotation equivariant models also achieve competitive performance, but with more parameters as well, when compared with RSESF. The poor performance of the $\mathcal{RST}$-CNN model, a UNet with jointly rotation-scale equivariant convolutional layers, suggests that has too few filters to learn adequate features. By tripling the number of filters we create $\mathcal{RST}$-CNN$^+$ with improved performance, but at the cost of tripling the amount of GPU required for training. This memory overhead is expensive; thus a trade-off between performance and computation resources needs to be considered.

**OOD testing.** In terms of OOD testing, which is designed to evaluate models' generalization capacity to rotation and scale variations, the proposed RSESF demonstrates the best performance on two histopathology datasets, as shown in Table 1. The $\mathcal{RST}$-CNN$^+$ outperforms the standard CNN, but the performance gain is limited when compared with RSESF, although it has slightly more parameters than RSESF. The performance of SEUNet ranks second and third on GlaS and CRAG datasets, respectively. This justifies the design choice that scale equivariant filters with learnable $\sigma$s are less sensitive to scale variation. By absorbing rotation equivariance into SEUNet, the mIoU of RSESF on GlaS and CRAG datasets are further boosted by 1.69 and 1.85 points. A statistical significance analysis on OOD testing results is provided in Appendix E.

**Prediction visualization.** In order to get a sense of how the RSESF demonstrates better generalization performance on OOD testing, we pick up an image from the test set of texture dataset and randomly rotate and re-scale the image for testing. As shown in Figure 2, E(2)CNN and H-Nets show stabilized prediction to rotation variations, but they fail to generalize accurately to up-scaled images. Although the H-Nets yields the highest mIoU score on the selected texture mosaics and OOD testing, RSESF shows the most robust performance for joint rotation and scale variations. The reason why H-Nets demonstrates better performance than RSESF is that the textures used in our experiments have specific orientation characteristic (see Figure 4), and when the model is trained on these textures without rotation augmentation, it causes the filters to respond only to features in a specific orientation. The circular harmonics used in the filter parameterization lead to full rotational equivariance, thus enabling H-Nets to capture orientation features in a fine-grained way. Although RDCF, $\mathcal{RST}$-CNN and $\mathcal{RST}$-CNN$^+$ also have 8 rotation channels, their robustness is manifest only for 4 rotation angles. Some segmentation maps are provided in Appendix H, Figure 5 and Figure 6.

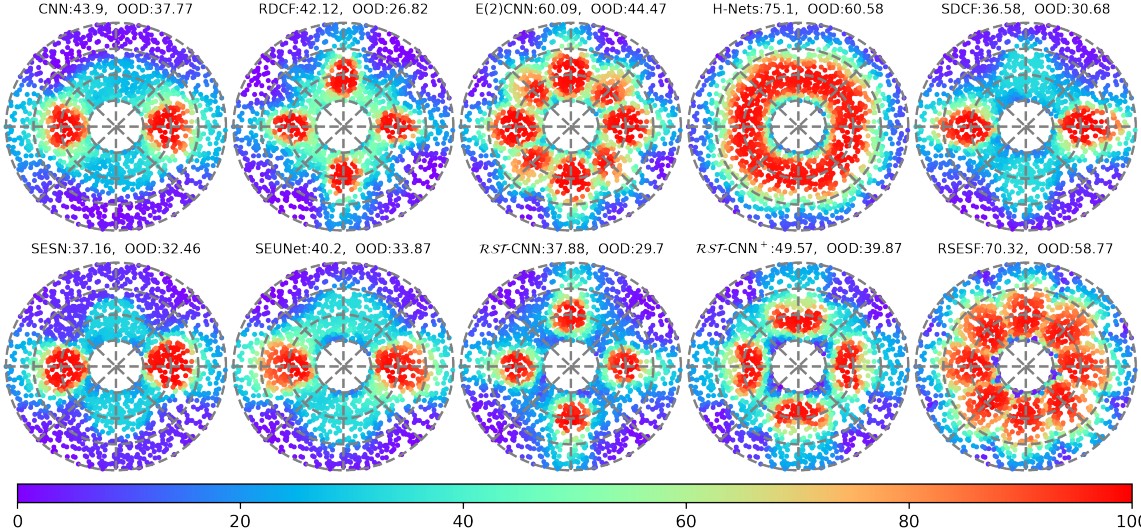

Figure 2: Polar scatter plots show mIoU scores for texture mosaics as a function of the orientation and scale of an input image. 2000 texture mosaics are generated through randomly rotating (by an angle uniformly distributed on $[0, 2\pi)$ and re-scaling (by a factor uniformly distributed on $[0.5, 2]$) the original mosaic. The radii of the four rings from inside to outside are 0.5, 1, 1.5 and 2, which represent the corresponding scaling factors. The mIoU of each prediction is indicated with colors. The averaged mIoU over 2000 predictions and that for the OOD testing on a dataset (Appendix C) of 160 texture mosaics that are rotated and re-scaled are shown on top of each plot respectively.

## 5. Conclusion

In this paper, we propose the RSESF, which can generalize convolutional neural networks to segment images presented in scales and orientations that do not exist in training samples. To this end, we parameterize filters by linearly combining groups of Gaussian derivative filters, within each one of the filters there is an additional rotation and scale channel to guarantee rotation and scale equivariance. The scale parameters are set to be both trainable yet cover disjoint ranges. Therefore scale equivariance is achieved and specific scale preference can be found for different datasets. The rotation channel can have $R$ filters of different orientations, spanning over $360°$ with an interval of $\frac{2\pi}{R}$ to guarantee rotation equivariance. Models with RSESF filters can be trained in a memory-efficient way, as the nature of decoupled equivariant convolution gives the model flexibility of training on one orientation but inference in multiple orientations. We also confirm experimentally that the RSESF achieves higher sample efficiency, when compared with normal CNNs. Although the filter constructed by using steerable basis filters can achieve continuous rotation equivariance in theory, it is infeasible to have infinite rotation channels in practice. This means RSESF is restricted to discrete rotations. In the future, we will explore generalizing discrete rotation angles to continuous rotation angles while making the model scale equivariant.

## Acknowledgement

The authors acknowledge the use of the IRIDIS High-Performance Computing Facility, and associated support services at the University of Southampton, in the completion of this work. The first author (Yilong Yang) is supported by China Scholarship Council under Grant No. 201906310150.

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

## Appendix A. The Second Order Directional Filter Basis

Here we derive the filter basis of order 2, as the highest order of the Gaussian derivative used in our experiments is 2. Denoting $z = G(x,y;\sigma)$ and $z^* = G_\theta^1(x,y;\sigma)$, then the second order directional derivative of $z$ with respect to angle $\theta$ can be calculated by:

$$
\begin{aligned}
G_\theta^2(x,y;\sigma) &= (\cos\theta,\ \sin\theta) \cdot (\frac{\partial z^*}{\partial x},\ \frac{\partial z^*}{\partial y}) \\
&= \cos\theta\frac{\partial z^*}{\partial x} + \sin\theta\frac{\partial z^*}{\partial y} \\
&= \cos\theta\frac{\partial G_\theta^1(x,y;\sigma)}{\partial x} + \sin\theta\frac{\partial G_\theta^1(x,y;\sigma)}{\partial y} \\
&= \cos\theta(\cos\theta\frac{\partial^2 G(x,y;\sigma)}{\partial x^2} + \sin\theta\frac{\partial^2 G(x,y;\sigma)}{\partial x\partial y}) \\
&\quad + \sin\theta(\cos\theta\frac{\partial^2 G(x,y;\sigma)}{\partial y\partial x} + \sin\theta\frac{\partial^2 G(x,y;\sigma)}{\partial y^2}) \\
&= \cos^2\theta\frac{\partial^2 G(x,y;\sigma)}{\partial x^2} + 2\cos\theta\sin\theta\frac{\partial^2 G(x,y;\sigma)}{\partial y\partial x} + \sin^2\theta\frac{\partial^2 G(x,y;\sigma)}{\partial y^2} \\
&= (\cos\theta\partial_x + \sin\theta\partial_y)^2 G(x,y;\sigma)
\end{aligned}
\tag{8}
$$

## Appendix B. Schematic of Constructing RSESF Filters

Here we provide a schematic representation in Figure 3 to illustrate the process of linearly combining Gaussian derivative filters.

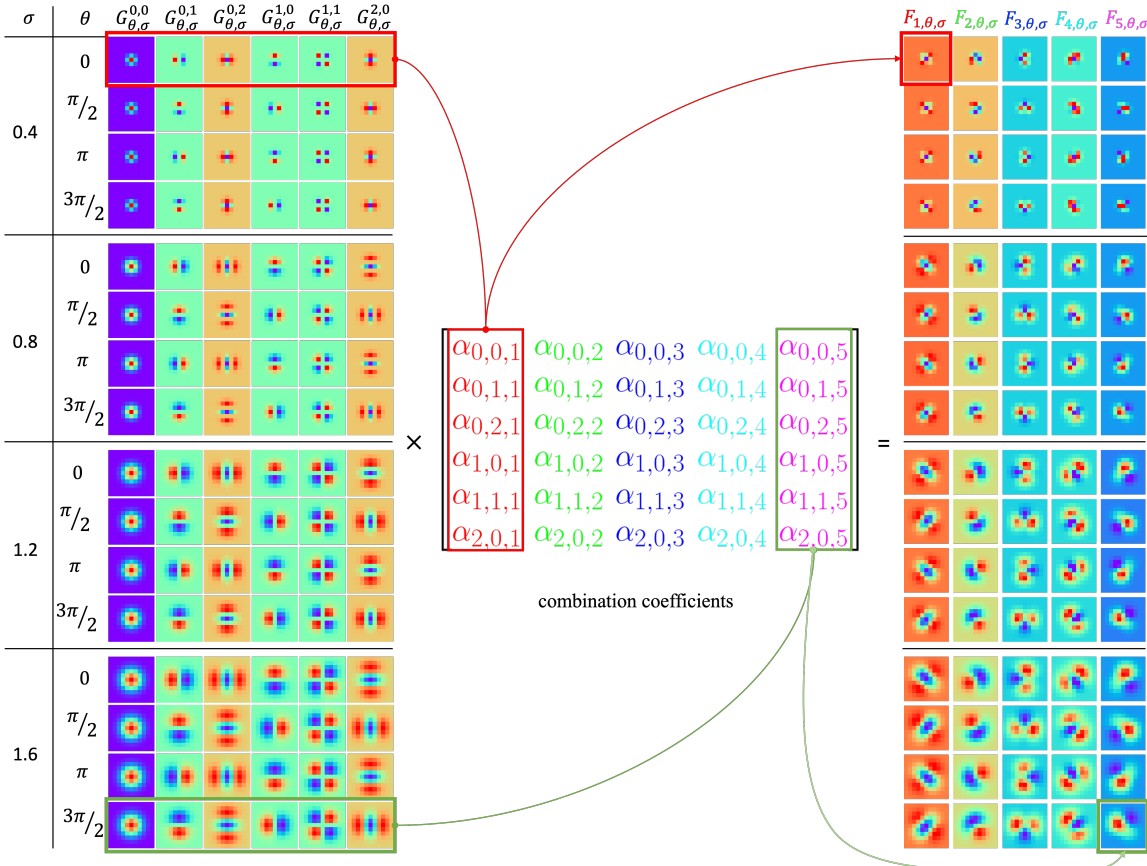

Figure 3: Here we depict the construction process of an RSESF filter (with 4 scale channels, 4 rotation channels, 1 input channel and 5 output channels) as matrix multiplication. The highest order of Gaussian derivative used here is 2. **Left.** Gaussian derivative filter basis $G_{\theta\sigma}^{i,j}$, parameterized by different $\theta$ and $\sigma$ values from top to bottom, and different orders $i$, $j$ (with respect to $\theta$ and $\theta + \frac{\pi}{2}$ respectively) from left to right; **Middle.** Learnable linear combination coefficients $\alpha_{i,j,c}$, where $c$ is the index of output channel. Each column vector is shared between scales, and it determines the shape of the final constructed filter. **Right.** The constructed multi-scale, multi-orientation filters (filters vary in shape from left to right, as learnable coefficients (column vectors) involved in the calculation are different). The red/green lines that connect the Gaussian derivative bases (left), combination coefficients (middle), and constructed filter (right) show the process of equation (3), i.e., the dot product between the Gaussian derivative bases and combination coefficients produces the convolutional filter. Note, for visualization purposes, we set $\sigma$ values to $\{0.4, 0.8, 1.2, 1.6\}$. However, they are allowed to be tuned in disjoint ranges as described in equation (6).

## Appendix C. Dataset Details

**CRAG dataset**. The colorectal adenocarcinoma gland (CRAG) dataset was originally used in (Awan et al., 2017); it contains a total of 213 Hematoxylin and Eosin images taken from 38 WSIs scanned with an Omnyx VL120 scanner under $20\times$ objective magnification). All images are mostly of size $1512\times1516$ pixels. The dataset is split into 173 training images and 40 test images. We resize each image to a resolution of $1024\times1024$ and then crop it into four patches with a resolution of $512\times512$ for all our experiments. Therefore, there are 692 patches in the training set and 160 patches in the test set. We then further split the training set into 552 images for training and 140 images as the validation set.

**GlaS dataset** The Gland Segmentation dataset (Sirinukunwattana et al., 2017) contains a total of 165 images ($20\times$ objective magnification) which are split into 85 images for training and 80 for testing. We crop four corners with the size of $512\times512$ from each image, resulting in 360 patches (from training images) and 320 patches (from test images). We further split the 360 image patches into a validation set (72 patches) and a training set (288 patches).

**Texture dataset.** Alongside evaluating models on histopathology datasets that have built-in rotation symmetry properties, we create a synthetic texture segmentation dataset made from directional texture images to analyze the effectiveness of the proposed method on out-of-distribution testing. In detail, we customize masks to each contain five regions using the online Prague Texture Segmentation Data Generator[1]. Then five types of pure texture images (160 unique texture patches per class) from the Kylberg Texture Dataset (the without rotation version) (Kylberg, 2011) are selected to create mixed mosaic images with boundaries aligned with mask shapes. Note that for each type of texture, all of the 160 pure texture patches are presented in only one orientation. We split 720 mosaic-mask pairs into 112 for validation, 448 for training and 160 for testing. Figure 4 shows examples of pure texture images and a synthetic texture mosaic with the mask.

**Creating test set for OOD testing.** For all datasets, to compare the generalization capability (the ability to segment images that are presented in unseen scales and orientations) of models, we randomly rotate and re-scale entire texture mosaics of the initial test set to create a new test set that has more orientation and scale variations for out-of-distribution testing.

---

1. https://mosaic.utia.cas.cz/

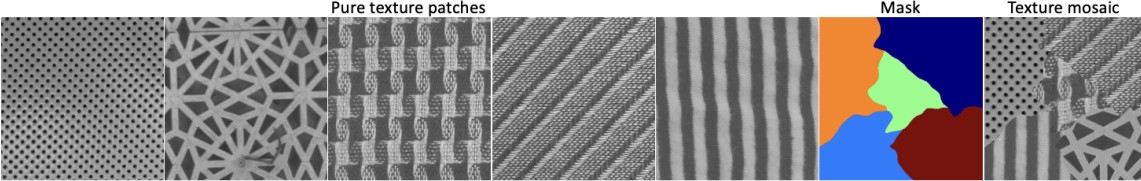

Figure 4: Pure texture images and generated texture mosaic with the corresponding mask.

## Appendix D. Model Settings and Implementation Details

For the UNet with norm CNN layers, the number of convolutional channels at each depth are 64, 128, 256, 512, 1024. For a fair comparison, we keep the total number of channels of equivariant UNet variants in line with the norm UNet. In detail, we split convolutional channels into $N_f$ filter channels, $k$ scale channels, $R$ rotation channels, and guarantee that $N_f \times k \times R$ is the same for all models. The value of $N_f$, $k$ and $R$ of each model are summarized in Table 1. For SEUNet and RSESF, we set the highest order of the Gaussian derivative to be 2. We carefully set the scale factors for SESN and SDCF, so that their receptive field sizes are in consistent with SEUNet and RSESF. For GlaS and CRAG datasets, the colour normalization method proposed in (Vahadane et al., 2016) is used to remove stain colour variation, before training. All models are trained on images at the original orientation and scale, only randomly horizontal and vertical flip augmentation is used in training.

We implement the conventional UNet model, SEUNet, and our RSESF. We use the e2cnn library[2] to build E(2)CNN and H-Nets. For RDCF, SDCF, SESN and $\mathcal{RST}$-CNN, the code associated with Gao et al. (2022) is used to build equivariant layers. All Models are implemented in Pytorch (Paszke et al., 2019) and trained on one NVIDIA RTX 8000 GPU (45GB memory) using the Adam optimizer (Kingma and Ba, 2014) to minimize the cross-entropy loss. To fully utilize the GPU memory for efficient training, we set the batch size to 6 for $\mathcal{RST}$-CNN$^+$ and 16 for all the other models. The learning rate, training epochs, and weight decay coefficient are optimized by Random Search (Bergstra and Bengio, 2012).

## Appendix E. Statistical Significance

We calculate the P-value of the mIoU scores (OOD testing results) between RSESF and other models and report them in Table 2 to show the statistical significance. As seen from the table, there is a statistically significant difference (p-value $<0.05$) between RSESF and other compared methods.

## Appendix F. Ablation Study

### F.1. Dimension Reduction at Rotation Channel

In section 3.4, we mentioned that the feature maps of the last layer from each scale group is max-pooled over rotation channels to be matched with the following $1 \times 1$ convolution layer

---

2. https://github.com/QUVA-Lab/e2cnn

Table 2: P-values of models that are trained without rotation and scale augmentation and evaluated on augmented test sets.

| DataSet | CNN | RDCF | E(2)CNN | H-Net | SDCF | SESN | SEUNet | $\mathcal{RST}$-CNN$^+$ |
|---------|------|------|---------|-------|------|------|--------|------------------------|
| GlaS | <0.001 | <0.001 | <0.001 | <0.001 | 0.039 | 0.010 | 0.018 | <0.001 |
| CRAG | <0.001 | <0.001 | 0.003 | <0.001 | <0.001 | <0.001 | 0.026 | <0.001 |
| Texture | <0.001 | <0.001 | 0.005 | 0.019 | <0.001 | <0.001 | <0.001 | <0.001 |

Table 3: Comparison of the segmentation performance when different channel squeeze strategies is applied on the last layer.

| Channel Squeeze at | In-Distribution | | | Out-of-Distribution | | |
|---|---|---|---|---|---|---|
| the Last Layer | GlaS | CRAG | Texture | GlaS | CRAG | Texture |
| Pooling over $R$ | 67.29 | 88.98 | 63.87 | 60.60 | **82.15** | 33.90 |
| Selecting $R^*$ over $C^l$ | 67.98 | **89.73** | **96.10** | **60.72** | 81.14 | **58.77** |
| Selecting $r^c$ for each $c$ | **68.10** | 89.49 | 73.31 | 60.71 | 81.67 | 35.76 |

for channel squeeze and prediction generation. Here, we explore other dimension reduction methods and summarize their performance in Table 3. Given the feature map $f_k^L$ that is computed by the last layer of the $k^{th}$ scale group, the following strategies are used to transfer $f_k^L \in \mathbb{R}^{C^L \times R \times H \times W}$ to $f_k^L \in \mathbb{R}^{C^L \times H \times W}$:

1. **Max-pooling $f_k^L$ over rotation dimension.** $f_k^L \in \mathbb{R}^{C^L \times R \times H \times W}$, $k = \{1, \cdots, \gamma\}$ has components $f_{k,r}^L$, $r \in \{1, \cdots, R\}$ for each pixel, then the maximum value over rotation channels is retained, i.e., $(f_k^L)^{\max} = \max\limits_{1 \leq r \leq R} f_{k,r}^L$.

2. **Selecting a unified rotation channel $R^*$ for all pixels over $C^L$ filter channels.** We sum $f_k^L$ along spatial and filter channel dimensions, the resultant tensor thus reflects the overall activation magnitude of each orientation. Then the rotation channel that has the largest activation value is selected and denoted as $R^*$:

$$R^* = \arg \max_{1 \leq r \leq R} \sum_{x=1}^{H} \sum_{y=1}^{W} \sum_{c=1}^{C^L} f_{k,c,r,x,y}^L, \tag{9}$$

   then for every pixel at each filter channel we obtain $f_{k,c,x,y}^L \leftarrow f_{k,c,R^*,x,y}^L$.

3. **Selecting a specific rotation channel $r_k^c$ for all pixels at each filter channel $c \in \{1, \cdots, C^L\}$.** We first sum $f_k^L$ along spatial dimension, the resultant tensor thus reflects the overall activation magnitude of each rotation channel $r$ and each filter channel $c$. Then the rotation channel with the highest mean activation values in filter channel $c$ is selected and denoted as $r_k^c$:

$$r_k^c = \arg \max_{1 \leq r \leq R} \sum_{x=1}^{H} \sum_{y=1}^{W} f_{k,c,r,x,y}^L, \tag{10}$$

   then for every pixel at each filter channel $c$ we obtain $f_{k,c,x,y}^L \leftarrow f_{k,c,r_k^c,x,y}^L$.

As shown in Table 3, all three strategies yield close performance on CRAG and GlaS datasets, but the second strategy outperforms others significantly on the texture dataset. We think the reason originates from the characteristic of datasets. Within each mask boundary, the textures of the cells and tissues appear in all orientations in the histology datasets.

Table 4: Comparison of the segmentation performance when a different number of rotation channels is applied during inference. R=1 is the original trained RSESF model.

| DataSet | Number of Rotation Channels (R) | | | | | |
|---------|------|------|------|------|------|------|
|         | 1    | 2    | 4    | 6    | 8    | 10   |
| GlaS    | 58.91 | 59.35 | 59.85 | 59.74 | **60.72** | 59.94 |
| CRAG    | 80.32 | 77.30 | 78.05 | 78.55 | **81.14** | 78.58 |
| Texture | 33.87 | 34.55 | 40.72 | 52.67 | 58.77 | **61.15** |

This is not so in texture mosaics, where each texture appears in only one orientation in the training set. Max-pooling over rotation channels mixes up orientation information of textures in a mosaic thus leading to incorrect prediction. Similarly, selecting specific rotation channel $r_k^c$ for each filter channel $c$ may also mix up orientation information, since $r_k^{c'} \equiv r_k^{c''}$ is not guaranteed, where $c'$, $c''$ denotes different filter channels. Selecting a unified rotation channel $R^*$ over all $C^L$ filter channels achieves the best performance on the texture dataset is reasonable since it does not mix up orientation information between filter channels.

**F.2. Flexibly Setting R While Inferencing**

In section 3.4, we demonstrate that RSESF possesses the flexibility of being trained with only one rotation channel but introducing other rotation channels during inference as needed to reduce the model's orientation sensitivity. Here we set different $R$ values to an RSESF model that is trained with one rotation channel (no rotation and scale augmentation is present during training) to create other 5 models, and then evaluate these models on randomly rotated and re-scaled test sets (OOD test set described in Appendix C). The dimension reduction strategy adopted in these experiments is "Selecting $R^*$ over $C^l$". For an RSESF model with $R$ rotation channels, the angular spacing between adjacent channels is $\frac{2\pi}{R}$. Increasing the number of rotation channels from 1 to $R$ also increases the amount of GPU memory by a factor of $R$. As seen from Table 4, increasing the number of rotation channels leads to the improvement of segmentation performance, when evaluated on the texture dataset. This is as expected since texture mosaics have specific directional texture patterns, an RSESF with fewer rotation channels is not able to pick up features that are presented in an angle it does not possess, therefore leading to inferior segmentation. But it is worth mentioning that an RSESF model (mIoU=61.54) with 10 rotation channels outperforms H-Nets (mIoU=60.58, reported in Figure 2), on the OOD testing criteria, even though the RSESF is not continuous rotation equivariant. We think the reason is that RSESF is jointly rotation-scale equivariant, therefore it can segment more precisely than H-Nets, when the test image is not just rotated, but also re-scaled. Examples shown in Figure 6 support this statement.

For CRAG and GlaS datasets, however, due to the inherent rotation symmetry of histopathology images, the relationship between the number of rotation channels and segmentation performance is not as straightforward as it is on the texture dataset. In histopathology images, texture patterns are already presented in arbitrary orientations during training.

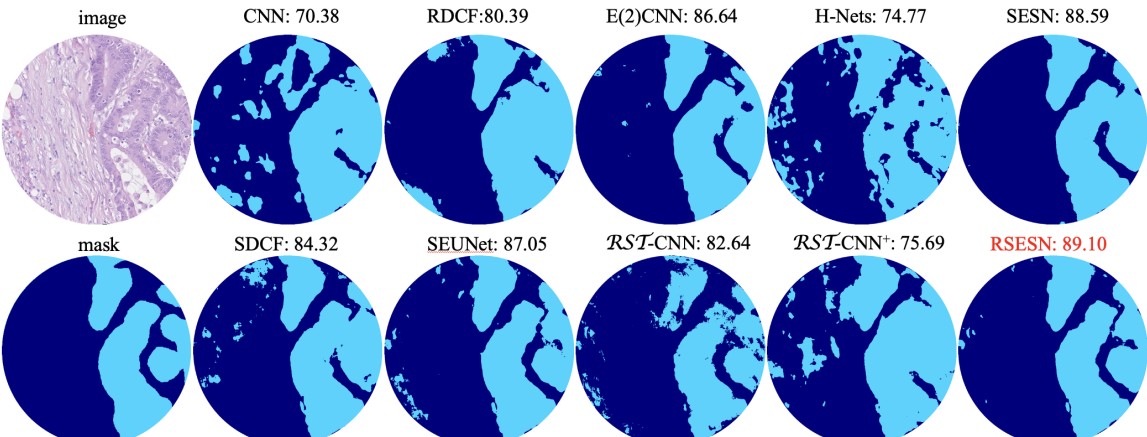

Figure 5: Images, masks, and predictions generated by models. The mIoU score of each prediction is shown on top of the segmentation map. The rotation angle and scaling factor of the selected histopathology image are $[r = 184°, s = 1.97]$. Light blue and dark blue represent gland and non-gland, respectively.

Changing the number of rotation channels can vary a model's performance, but one can not draw a conclusion that more rotation channels lead to better performance. However, a validation set can be used to search for the optimal $R$.

## Appendix G. Prediction Visualization

In Figure 5, we select a randomly rotated and re-scaled image from the test set of CRAG for visualization. Some segmentation maps of texture mosaics that are presented in different scales and/or orientations are shown in Figure 6.

Figure 6 provides some visual clues of how different type of models demonstrate their superiority on different versions of test images. As seen from Figure 6, when the test mosaic is rotated by 58° (without re-scaling it), E(2)CNN, H-Nets and RSESF can remain relatively high prediction accuracy (80.90 v.s. 80.22 v.s. 74.82). The reason that H-Nets outperforms RSESF is that H-Nets has continuous rotation equivariance property while the orientation of 58° is not encoded in RSESF. When the test mosaic is re-scaled (without rotating it), all of the scale equivariant models, SDCF (82.46), SESN (82.44) and SEUNet (92.85) show more accurate segmentation. When the test mosaic is rotated and re-scaled simultaneously, although the performance of all models degrades, the RSESF is the one that demonstrates the best performance.

## Appendix H. Additional Experiments

Apart from experiments conducted in section 4, here we design experiments in which UNets with norm CNN filter and RSESF are trained either with or without rotation augmentation to demonstrate the superiority of RSESF on sample efficiency.

**Datasets.** We use the Texture dataset for model training and evaluation.

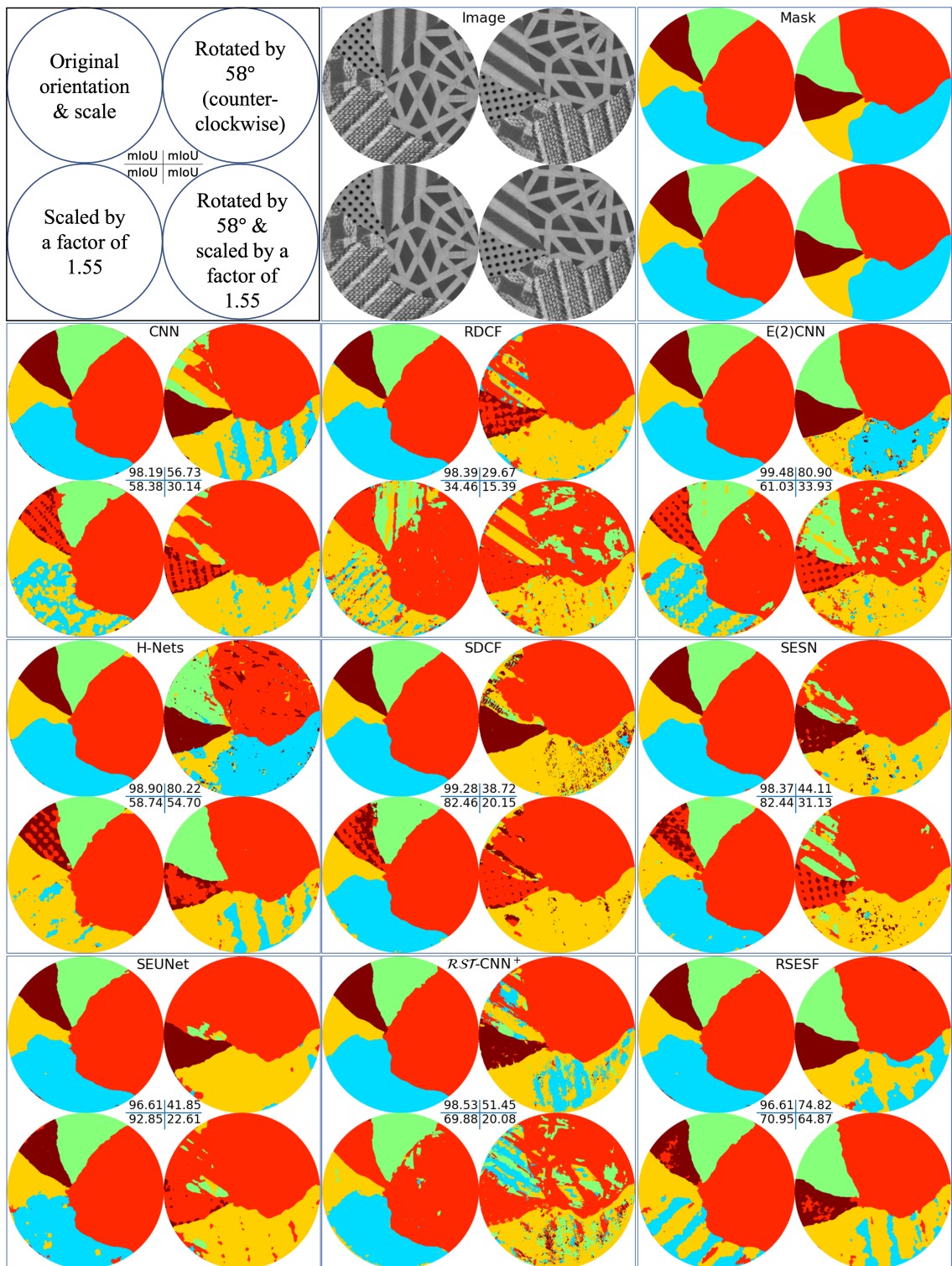

Figure 6: The mIoU scores are reported at the center of each rectangular block.

1) The training set 1 (the initial training set introduced in Appendix C), in which 560 texture mosaics are presented in one particular orientation, is used for model training under the setting of without rotation augmentation.

2) The training set 2 is the rotation-augmented version of training set 1, where each texture mosaic is presented in 6 angles ($0°$, $60°$, $120°$, $180°$, $240°$, $300°$). Therefore there are 3360 texture mosaics in total. Training set 2 is used for model training under the setting of with rotation augmentation.

3) For evaluation, we re-scale and rotate the texture mosaics of the initial texture test set (introduced in Appendix C) by 36 angles $\{\frac{2\pi r}{36}\}_{r=1}^{36}$ and 9 scaling factors $\{\frac{\sqrt[4]{2}^s}{2}\}_{s=0}^{8}$. Therefore there are 324 subsets of mosaics, consist of 51840 ($160 \times 9 \times 36$) texture mosaics in total.

**Model settings.** We create an UNet with RSESF filters that has 3 scale groups. By constraint the upper and lower bounds of $\sigma_k^l$ using equation (6), we therefore constraint the size of filters at each group to be $[5 \times 5]$, $[7 \times 7]$ and $[9 \times 9]$, for every layer. For fair comparison, we match the size of norm CNN filters with that of each scale group of RSESF filters. In detail, we create CNN_$[5 \times 5]$, CNN_$[7 \times 7]$ and CNN_$[9 \times 9]$, where CNN_$[k \times k]$ means that the size of filter is set to be $k \times k$ for every layer of the UNet. The number of filters is set to be 60, 120, 240, 480 and 960, at each depth of the CNN-based UNet. For RSESF-based UNet, the number of filter channels are divided by 3. When evaluation, we generate segmentation map for each scale groups and report their performance separately.

**Results.** As shown in Figure 7, when models are trained without rotation augmentation, CNNs only show competitive performance on mosaics that are presented at the same orientation as training mosaics and a large fluctuation can be see from Figure 7(a). In contrast, as shown in Figure 7(c), RSESF demonstrates relative stable prediction over orientations. When models are trained with rotation augmentation (with 5 times more training samples), both performance and robustness of CNNs are greatly improved, while some fluctuations between orientations still remain. The overall performance of RSESF-based UNet also benefits from rotation augmentation. It is worth mentioning that RSESF-based UNet trained on 560 mosaics achieves close performance to CNN-based UNet trained on 3360 mosaics (CNN_$[7 \times 7]$ vs. $\sigma_{2\_}[7 \times 7]$ and CNN_$[9 \times 9]$ vs. $\sigma_{3\_}[9 \times 9]$). In addition, the number of filter channels of RSESF is only one-third of that of CNN. This comparison highlights the higher sample efficiency of RSESF.

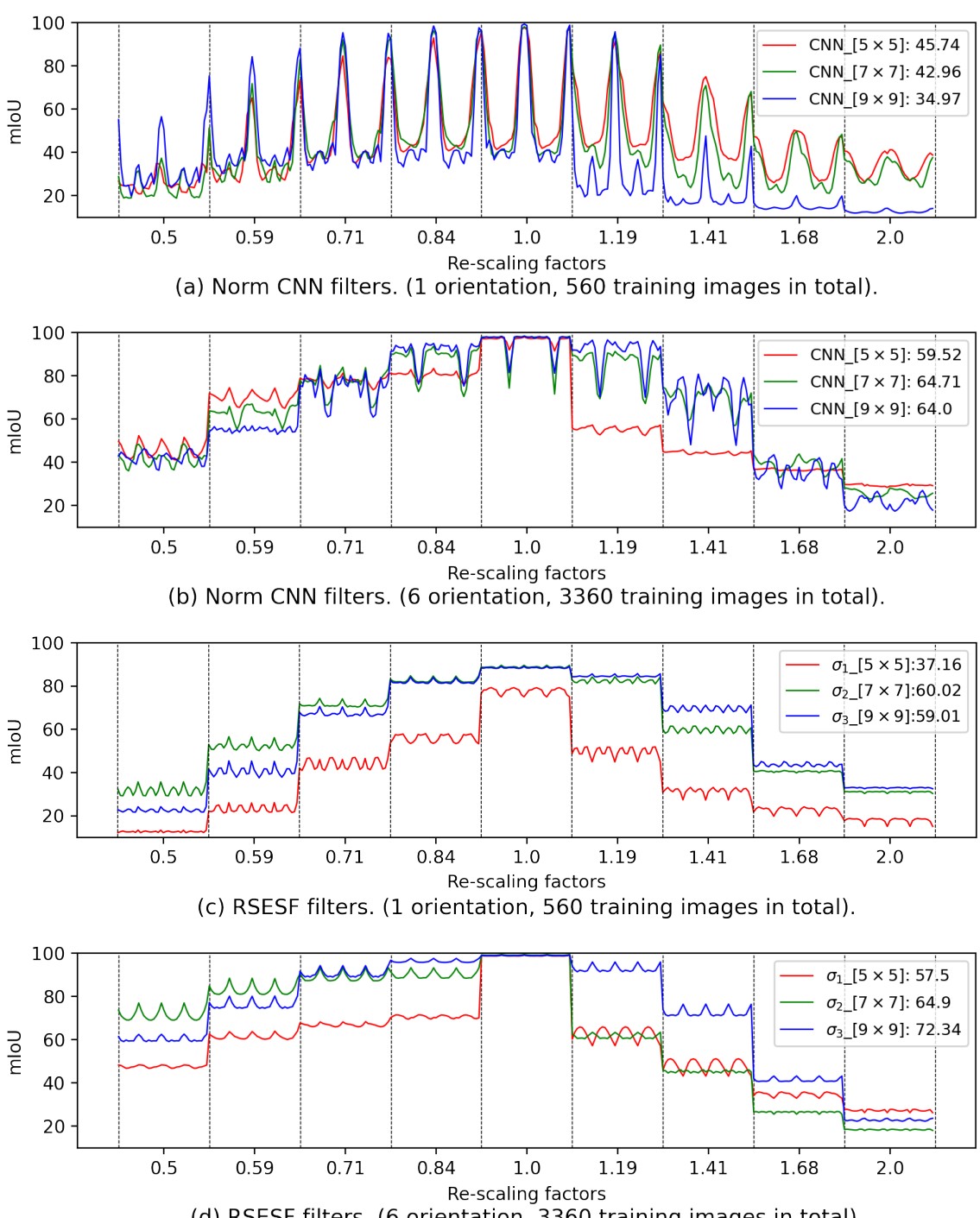

(a) Norm CNN filters. (1 orientation, 560 training images in total).

(b) Norm CNN filters. (6 orientation, 3360 training images in total).

(c) RSESF filters. (1 orientation, 560 training images in total).

(d) RSESF filters. (6 orientation, 3360 training images in total).

Figure 7: mIoU score measured on multi-scale multi-orientation texture mosaics. Within each column (separated by vertical dashed lines), images are re-scaled by the same factor shown in the x-axis and rotated by 36 different angles (no show, 0° centred in each column). The averaged mIoU over all orientations and scales are reported on legends.

