# OpenReview forum: "Rotation-Scale Equivariant Steerable Filters"
_MIDL.io/2023/Conference — MIDL 2023 Poster_

### Official Review · Reviewer_EN5C · 2023-02-03

**Confidence:** 4
**Preliminary Rating:** 5
**Recommendation:** Best Paper Award, Oral

**Summary:**

This is a great work on steerable filters for both rotational and scale equivariance. The authors demonstrate both in their technical description and experimental validation with great detail that the proposed implementation (which linearly combines Gaussian filters) achieves high accuracy and low memory demand for histopathology image analysis compared to state-of-the-art.

**Strengths:**

- The method is novel and of high interest for the community, in particular the flexibility to achieve different memory savings in training and at inference time is valuable for practical implementation
- The paper is very well written and nicely illustrated
- Both in-domain and out-of-domain scenarios are considered for segmentation in two histopathology datasets with a U-Net backbone
- The experimental evaluation is comprehensive and significant improvements over all methods in OOD (H-nets comes close) are found


**Weaknesses:**

- not necessarily a weakness but maybe a misunderstanding: The method only considers scale-invariance at the first convolution (and afterwards filters across scale-groups are treated in parallel), which could indicate that an input augmentation (or ensemble across input scales) can have a similar effect, right? This question is also related to the finding (Appendix E) that selecting a "unified rotation angle R*" yields the best overall segmentation performance. Is this due to the memory-efficient training that avoids channel mixing or can this be explained based on the visual features of the data (e.g. "texture")?
- to my understanding only flips and translations are considered as augmentation (see Appendix C) for the baseline CNN-UNet: this is of course not ideal when aiming to segment images that can be found in very different orientations

**Deanonymize Review:**

yes

**Detailed Comments:**

I think some further discussion on the number R of discrete rotations during inference and its impact on accuracy could be good.

**Paper Type:**

methodological development

**Questions To Address In The Rebuttal:**

I would already recommend acceptance of the paper in its current version, but might further upgrade my score (to best paper candidate) if the authors could add the "better" baseline of a CNN-UNet trained with rotation/scale input augmentation (even if it performs best). I realise there is some experiment in that direction in the last Appendix on the synthetic texture data, but this could be integrated for the real datasets in the main body.

Final justification:
I would like to re-confirm my initial strong acceptance vote and thank the authors for providing additional insights and clarifications. This alleviates in my opinion any remaining concerns and I believe the nature of the overall mixed review scores stem from a very strict definition of relevance to the community of one co-reviewer - which should have been avoided according to the guidelines. A minor point (taken from the discussions with Reviewer d7ig): I also agree with the reviewer that online / on-the-fly augmentations are preferable in medical imaging as their computation cost is usually negligible compared to the forward and backward passes through the CNN.

---

### Official Review · Reviewer_d7ig · 2023-02-03

**Confidence:** 4
**Preliminary Rating:** 1

**Summary:**

This paper introduces a joint integration of rotation and scale equivariance into CNNs.
It is motivated by the arbitrary orientation of cells in histology images and the different resolutions.

Models are evaluated on 3 datasets: 2 gland segmentation datasets and one synthetic  texture dataset.
The proposed approach is compared with other invariance methods from the literature.


**Strengths:**

The paper is relatively easy to read (although notations and equations could be polished and simplified). The method is simple, and the experiments are conducted on three public segmentation datasets.


**Weaknesses:**

For different resolutions/magnifications of histopathology images, it is rather a multi-scale approach (with a large literature for multi-scale in histopathology) that is relevant, not an invariance. The scale has a meaning (correspondence between  pixel spacing and physical dimension), and the information about the resolution can be passed to the algorithm.
See e.g.:
Graziani, Mara, et al. "On the scale invariance in state of the art CNNs trained on ImageNet." Machine Learning and Knowledge Extraction 3.2 (2021): 374-391.
I do not see a clinical scenario for which a scale invariant algorithm would be required. The only application I can think of is when the magnification is unknown, maybe using images from the literature for which the magnification would not be reported
See e.g.:
Otálora, Sebastian, et al. "Image magnification regression using densenet for exploiting histopathology open access content." Computational Pathology and Ophthalmic Medical Image Analysis: First International Workshop, COMPAY 2018, and 5th International Workshop, OMIA 2018, Held in Conjunction with MICCAI 2018, Granada, Spain, September 16-20, 2018, Proceedings 5. Springer International Publishing, 2018.

The OOD experiment is not relevant. The other models are trained for a given resolution, we do not expect them to generalize to new resolutions. A multi-scale approach should be used as a comparison, or different models trained for different resolutions.



**Deanonymize Review:**

no

**Detailed Comments:**

More detailed comments:
- The filters are non-trainable: copies of filters that are linear combinations of Gaussian filters. A more similar work and missing literature seems to be the scattering work of Mallat, e.g.
Bruna, Joan, and Stéphane Mallat. "Invariant scattering convolution networks." IEEE transactions on pattern analysis and machine intelligence 35.8 (2013): 1872-1886.

- The work is also not put in the context of the strong group theory in CNNs:
Cohen, Taco S., Mario Geiger, and Maurice Weiler. "A general theory of equivariant cnns on homogeneous spaces." Advances in neural information processing systems 32 (2019).

- Models should also be trained with augmentation in addition, for fair comparison.

- “One can use data augmentation for better generalization, but this is at the cost of increasing the number of training samples.” This is misleading, not increasing the number of training samples as generally thought of. And it is not a problem as such to increase with augmentation.


**Paper Type:**

methodological development

**Questions To Address In The Rebuttal:**

In the current form, I do not think the paper is relevant for the MIDL community, both due to the lack of novelty/inclusion within the existing literature, and the fact that scale invariance is not relevant for histology image analysis. These are the two major points that would require modifications, but it would become a completely different paper.

---

### Official Review · Reviewer_eum2 · 2023-02-05

**Confidence:** 4
**Preliminary Rating:** 4
**Recommendation:** Poster

**Summary:**

The paper proposes a new Rotation-Scale Equivariant formulation for CNNs that, as the name says, achieves generally better Rotation-Scale Equivariance than previous published methods with generally lower parameter counts. Results are somewhat mixed, in the sense that the proposed method is not better than competing methods for all experiments independently , but it appears to be overall the method with the best overall performance.

**Strengths:**

- Nice theoretical work with an interesting and novel (to my knowledge) filter formulation.
- Some time and effort was spent on making the method computationally and memory optimal
- They used 3 datasets and ran extensive metrics across these
- Authors compared the proposed method with a large collection of methods proposing different filter setups
- Results of fig 2 are very interesting and visually appealing

**Weaknesses:**

- Highly theoretical work where the are of application is an afterthought.
- Results are mixed bag, with Table 1 and Fig 2 showing different method performing better according to different criteria.
- There is too much of a focus on the number of parameters in Fig1 and through the text.

**Deanonymize Review:**

no

**Detailed Comments:**

Expanding on the weaknesses above:
- Highly theoretical work where the are of application is an afterthought. Would probably be better received in a non-medical venue. The notation of sections 3.2 and 3.3 is also hard to digest and deserves a pictorial representation of the linear combination of steered gaussian derivative filters as a way to translate the mathematics into an intuitive understanding.
- Results are mixed bag, with Table 1 and Fig 2 showing different method performing better according to different criteria. Little discussion is provided regarding the WHY of the results. For example, H-nets perform very well in Fig 2 (I would argue it is the best method) but then the performance in table 1 is somewhat poor. The E(2)CNN seems to be very close to the proposed method in fig 2 (slightly lower mIoU but same patter), and good ID testing performance in table 1, but poorer OOD performance. Why is this?
- There is too much of a focus on the number of parameters in Fig1 and through the text. As long as the method is computationally feasible, one should not strive for less parameters as an aim of itself. Most methods in Table 1 run on commodity hardware, so the numbers are almost irrelevant, unless authors can show that this allows for completely different model scaling.

**Paper Type:**

methodological development

**Questions To Address In The Rebuttal:**

I don't quite have questions per se, but I can convert the items above into questions:
- Why not submit the paper to a non-medical venue?
- Why do methods that perform well in Fig 2 (have good rotation and scale equivariance) don't seem to perform as well in terms of raw model accuracy?
- Why are the number of parameters important? There does not seem to be any correlation between parameters and performance.

---

### Official Review · Reviewer_m9VC · 2023-02-06

**Confidence:** 3
**Preliminary Rating:** 3
**Recommendation:** Poster

**Summary:**

This paper proposed rotation-scale equivariant steerable filters, which jointly integrate rotation and scale equivariance into CNNs for the digital histopathology imaging applications. The proposed methods were evaluated on two gland segmentation datasets. It was demonstrated that the proposed methods are superior to other competing methods with fewer learnable parameters.

**Strengths:**

Rotation and scale invariant filters are critical for the digital pathology images. Previous works only partially solved the problem by either rotation or scale invariance. This paper jointly integrated rotation and scale invariant filters by using a Gaussian derivative filter basis with scale and orientation parameters. This paper is well written and the authors derived the equivariance with the formulas in the methods section.

**Weaknesses:**

In the evaluation of the proposed model, the authors divided the data into training and testing for each dataset. Considering the small sample size, the models should be tested with cross-validation. I don’t know whether the authors used a separate validation set to optimize the hyperparameters.

**Deanonymize Review:**

no

**Detailed Comments:**

I understand that the authors tried to explain the derivation of equivariance of filters and how to construct those filters in the subsequent equations. These are helpful but it’s hard to follow all derivations due to too many superscripts and subscripts of symbols. For example, in equation (3), it might be better to omit the layer index ‘l’ and explain in the sentence that the equation is for a fixed layer ‘l’. Simplification of notations might be more readable.

**Paper Type:**

methodological development

**Questions To Address In The Rebuttal:**

- As explained in the weakness, have the authors used a validation dataset to optimize hyperparameters? How were the training parameters set?
- The quantitative results were shown in Table 1 with mIoU values listed. Is there any statistical significance among those results?

---

### Meta-Review · Area_Chair_YUCy · 2023-02-20

**Recommendation:** Accept (Poster)
**Confidence:** 4

**Metareview:**

The paper proposes rotation-scale equivariant steerable filters to integrate rotation and scale equivariance into CNNs for digital histopathology imaging applications. The paper is well-written, and the authors derived the equivariance with the formulas in the methods section. They used three datasets and ran extensive metrics across these, comparing the proposed method with a large collection of methods proposing different filter setups. Results are mixed, with Table 1 and Fig 2 showing different methods performing better according to different criteria. The evaluation would benefit from cross-validation instead of dividing data into training and testing for each dataset. The focus on the number of parameters in Fig1 and through the text is overemphasized, and a multi-scale approach might be more relevant for different resolutions/magnifications of histopathology images. The OOD experiment is not relevant. After the rebuttal, I think the work has some merits that can be presented at the conference.